# Research

ecology, evolution, genetics

bumblebee, colonization, diploid male production, genetic paradox of invasion, invasive species, population genetics

**Authors for correspondence:**
Ryan E. Brock
e-mail: ryan.brock@uea.ac.uk
Andrew F. G. Bourke
e-mail: a.bourke@uea.ac.uk

†These authors contributed equally to the study.

# No severe genetic bottleneck in a rapidly range-expanding bumblebee pollinator

Ryan E. Brock[1,†], Liam P. Crowther[1,†], David J. Wright[1,2], David S. Richardson[1], Claire Carvell[3], Martin I. Taylor[1] and Andrew F. G. Bourke[1]

[1]School of Biological Sciences, University of East Anglia, Norwich Research Park, Norwich, Norfolk NR4 7TJ, UK
[2]Earlham Institute, Norwich Research Park, Colney Lane, Norwich, Norfolk NR4 7UZ, UK
[3]UK Centre for Ecology and Hydrology, Maclean Building, Crowmarsh Gifford, Wallingford, Oxfordshire OX10 8BB, UK

REB, 0000-0003-2977-1370; LPC, 0000-0002-3004-0359; DSR, 0000-0001-7226-9074;
MIT, 0000-0002-3858-0712; AFGB, 0000-0001-5891-8816

Genetic bottlenecks can limit the success of populations colonizing new ranges. However, successful colonizations can occur despite bottlenecks, a phenomenon known as the genetic paradox of invasion. Eusocial Hymenoptera such as bumblebees (*Bombus* spp.) should be particularly vulnerable to genetic bottlenecks, since homozygosity at the sex-determining locus leads to costly diploid male production (DMP). The Tree Bumblebee (*Bombus hypnorum*) has rapidly colonized the UK since 2001 and has been highlighted as exemplifying the genetic paradox of invasion. Using microsatellite genotyping, combined with the first genetic estimates of DMP in UK *B. hypnorum*, we tested two alternative genetic hypotheses ('bottleneck' and 'gene flow' hypotheses) for *B. hypnorum*'s colonization of the UK. We found that the UK population has not undergone a recent severe genetic bottleneck and exhibits levels of genetic diversity falling between those of widespread and range-restricted *Bombus* species. Diploid males occurred in 15.4% of reared colonies, leading to an estimate of 21.5 alleles at the sex-determining locus. Overall, the findings show that this population is not bottlenecked, instead suggesting that it is experiencing continued gene flow from the continental European source population with only moderate loss of genetic diversity, and does not exemplify the genetic paradox of invasion.

## 1. Introduction

Colonization and invasion events involve changes in population size, with founding populations typically representing a subset of the source population. This reduction in population size creates a sampling effect on alleles, reducing genetic diversity [1]. Such a phenomenon, known as a genetic bottleneck, can lead to reduced adaptive potential [2], inbreeding and subsequent loss of heterozygosity [3] and stochastic increases in the frequency of deleterious alleles [4]. In turn, these processes may reduce the fitness of a founding population and thereby hinder its establishment and spread across new ranges. However, there are numerous examples of species that have successfully colonized new ranges after undergoing severe genetic bottlenecks during their initial introduction [5,6], a phenomenon known as the genetic paradox of invasion [7]. Hence, despite counter-cases [6,8], the relationship between colonization success and levels of genetic diversity in founding populations remains to be fully resolved.

Genetic bottlenecks are potentially even more harmful in the Hymenoptera (ants, bees, wasps and sawflies) due to single-locus complementary sex determination (*sl*-CSD). In *sl*-CSD, allelic diversity at a single locus, combined with haplodiploidy, determines an individual's sex [9]. Specifically, diploid individuals heterozygous at the sex-determining locus develop as females whereas haploid individuals hemizygous at the sex-determining locus develop as males. Under low genetic diversity and/or inbreeding, there will be an increasing frequency

of mating pairs sharing an allele at the sex-determining locus (matched mating). In such cases, 50% of diploid offspring produced will be homozygous at the sex-determining locus and therefore develop as diploid males [10].

Diploid male production (DMP) is costly since diploid males are inviable or sterile [11,12] or produce inviable or sterile triploid offspring [13,14]. These negative fitness impacts are exacerbated in colonies of eusocial Hymenoptera (all ants, bees and wasps with a worker caste), in which diploid males replace half of the potential workers but do not contribute to colony productivity [10]. Accordingly, DMP reduces colony founding success and productivity in both ants and bumblebees (*Bombus* spp.) [15–18]. In addition, a high frequency of DMP represents a strong indicator of a genetic bottleneck [19,20]. Despite this, eusocial Hymenoptera account for some of the most invasive species worldwide [21,22], with several cases occurring in which invasive populations have undergone severe bottlenecks [23–27]. For example, the yellow-legged Hornet (*Vespa velutina*) has successfully colonized much of southern Europe following a founding event inferred to involve a single multiply-mated queen [23].

Eusocial Hymenoptera perform essential services across natural and farmed ecosystems [28]. In particular, eusocial bees, including bumblebees and the honeybee *Apis mellifera*, represent some of the most important insect pollinators of food crops and wild plants [29], a role that is threatened by their widespread declines (e.g. [29–31]). However, at regional scales, some bee species are expanding their ranges, through either human transportation or natural colonization events [26,32–35]. Given concerns over maintaining bee populations, there is special interest in determining the relationship between ecological success and the genetic consequences of bottlenecks, mediated by *sl*-CSD, in range-expanding eusocial bee species [24,26].

The Tree Bumblebee (*B. hypnorum*) represents a highly successful range-expanding bumblebee. Having been initially recorded in southern England in 2001 [32], apparently as a natural colonist, it has rapidly increased in range and abundance to become one of the most widespread and common UK bumblebee species [36–38]. Historically, the range of *B. hypnorum* extends across continental Europe and Asia [32,39]. Hence, it seems likely that *B. hypnorum* arrived in the UK from the closest neighbouring area of its pre-2001 range, northern France [32,39].

A previous study suggested that *B. hypnorum* underwent a severe genetic bottleneck on its arrival in the UK, with male production in the first brood (indicative of DMP, as first broods are usually composed of workers alone) being observed in three of 13 colonies reared from field-collected queens [38]. These data were used to estimate that the sex-determining locus in the UK *B. hypnorum* population has four alleles and that the founding population consisted of one or two multiply mated queens [38]. Consequently, the successful establishment and spread of *B. hypnorum* in the UK despite an apparently severe genetic bottleneck has been cited as a prime example of the genetic paradox of invasion [5].

However, the previous work [38] did not confirm DMP genetically, nor account for facultative polyandry (multiple mating by queens) in *B. hypnorum* [37,40–43], potentially leading to an inaccurate estimate of allelic diversity at the sex-determining locus [44]. Recently, up to 11 alleles were found at neutral microsatellite loci in workers of a UK *B. hypnorum* population, in which queen mating frequency was

estimated at 1.7 mates per queen [37], suggesting a founding population of greater than two queens. Moreover, recording data show that *B. hypnorum* has expanded its range westwards within continental Europe across Germany and Belgium from the middle of the twentieth century [35,45,46], suggesting that colonization of the UK may represent part of an ongoing, large-scale range expansion in Europe. Hence, whether colonization of the UK by *B. hypnorum* truly exemplifies the genetic paradox of invasion is uncertain. This suggestion is also consistent with the findings of a recent RAD-seq study that showed similar levels of genetic diversity and no evidence of structuring between six UK *B. hypnorum* populations and one in northern France [47].

Therefore, we defined two contrasting genetic hypotheses to characterize the mode of colonization of the UK by *B. hypnorum*. Under the 'bottleneck hypothesis', a small number of individuals founded the entire UK population in a single, chance event [38]. This hypothesis predicts that the UK *B. hypnorum* population will show low genetic diversity, evidence of a recent severe genetic bottleneck (i.e. at the time of colonization) and high levels of DMP. By contrast, under the 'gene flow hypothesis', colonization of the UK by *B. hypnorum* represented part of an ongoing, large-scale westward range expansion, with a larger founding population and subsequent continued immigration from continental European populations [48]. This hypothesis predicts that the UK *B. hypnorum* population will show high genetic diversity, no recent severe genetic bottleneck and low levels of DMP.

Using a panel of previously characterized polymorphic microsatellite loci [37], we sought to discriminate between these two hypotheses and so establish whether *B. hypnorum*'s colonization of the UK truly represents a genetic paradox of invasion. To this end, within a representative UK population of *B. hypnorum*, we pursued two aims. First, we quantified genetic diversity in workers (using data from [37]) and males to perform the first test of whether the UK *B. hypnorum* population has undergone a severe genetic bottleneck relative to established UK bumblebee species. Second, because determining levels of DMP provides a powerful independent means of estimating levels of genetic diversity in eusocial Hymenoptera, we estimated the frequency of diploid males and allelic diversity at the sex-determining locus genetically for the first time in the UK *B. hypnorum* population.

## 2. Material and methods

### (a) Genetic diversity and bottleneck analyses

#### (i) Worker sample collection and genotyping

Workers from a population of *B. hypnorum* in Norwich, Norfolk, UK, were used to test for a recent reduction in effective population size (bottleneck), i.e. one occurring at the time of colonization [32]. A total of 675 *B. hypnorum* workers were sampled non-lethally [49] from a 2 km² area across two consecutive summers (2014: $n = 398$; 2015: $n = 277$). Of these, 645 workers (2014: $n = 375$; 2015: $n = 270$) were genotyped at up to 14 microsatellite loci (median = 11 loci), following methods described in detail in Crowther *et al.* [37] and in the electronic supplementary material (see also Crowther *et al.* [37] for original data).

#### (ii) Genetic diversity and bottleneck evaluation

All 645 worker genotypes from Crowther *et al.* [37] were used to estimate measures of genetic diversity, i.e. the number of alleles

per locus, mean allelic richness ($A_R$), and mean observed ($H_o$) and expected heterozygosity ($H_e$).

Using single workers randomly selected from each of 89 independent clusters among the worker genotypic data identified with COLONY v. 2 [50] (see the electronic supplementary material), two different methods were employed to test for a bottleneck in the *B. hypnorum* study population. The first was a sign test implemented in the programme BOTTLENECK 1.2.02 [51], in which an excess of expected heterozygosity over that expected under mutation–drift equilibrium suggests that a recent reduction in population size has occurred [52]. The second test calculated the M-ratio [53] across loci for the *B. hypnorum* study population. The M-ratio defines the ratio between allelic diversity and allele size range at a locus, with bottlenecks leading to the stochastic loss of rare alleles and a subsequent reduction of the M-ratio [53]. Under mutation–drift equilibrium, an M-ratio of less than 0.7 can be interpreted as evidence of a historical population reduction, a signal that may persist for more than 100 generations [53].

To provide a set of comparative M-ratios in established UK *Bombus* species, M-ratios were also calculated for UK populations of the five species, *B. hortorum*, *B. lapidarius*, *B. pascuorum*, *B. ruderatus* and *B. terrestris* [54]. These 'reference' *Bombus* species have not undergone range expansions within the UK, and, except for the scarce *B. ruderatus*, are common and widespread. Therefore, M-ratios calculated for these species provided null values against which the M-ratio calculated for *B. hypnorum* was compared. Hence, if *B. hypnorum* experienced a severe genetic bottleneck upon its colonization of the UK, it would be expected to exhibit a lower M-ratio than the reference *Bombus* species. Full details of both tests are in the electronic supplementary material.

## (b) Diploid male production and allelic diversity at the sex-determining locus

### (i) Male sample collection and genotyping

To estimate levels of DMP in *B. hypnorum*, males were sampled from two sources. First, 380 male pupae were sampled from 20 mature *B. hypnorum* colonies collected in the field in Norfolk and Suffolk, UK, over two consecutive years (2017: $n_{colonies} = 17$, $n_{male\ pupae} = 337$; 2018: $n_{colonies} = 3$, $n_{male\ pupae} = 43$). These 20 colonies each provided 7–24 randomly sampled male pupae for genotyping (electronic supplementary material, table S1).

Second, to allow for survivorship biases in field-collected nests (as detailed in the electronic supplementary material), adult males were sampled from *B. hypnorum* colonies reared from 107 queens collected from field sites in Surrey, Greater London, and Norfolk (Norwich), UK, during spring 2018 (electronic supplementary material, table S2). In total, 37 of the 107 field-collected queens reared at least one adult offspring (electronic supplementary material, table S3), with nine colonies producing only workers, six colonies producing only males and 22 colonies producing both workers and males. Of the 28 colonies that produced males, 12 were assigned as 'first-brood male' producers, which were defined as colonies in which either any males eclosed within one week of first worker eclosion ($n_{colonies} = 6$) or only males and no workers eclosed ($n_{colonies} = 6$) (mean [range] $n$ of first-brood males produced per colony = 2 [1–7]); the remaining 16 were assigned as 'late male' producers, which were defined as colonies in which all males eclosed later than one week after first worker eclosion (electronic supplementary material, figure S3 and table S3). From the 28 male producing colonies, a total of 232 adult males were sampled for genotyping (electronic supplementary material, tables S3–S5), comprising 25 first-brood males, which were defined as males that either eclosed within

one week of first worker eclosion or were produced by colonies producing no workers, and 207 late males, which were defined as males that eclosed later than one week after first worker eclosion.

All sampled males ($n = 612$, i.e. 380 pupal males from field-collected nests plus 232 adult males from colonies reared from field-collected queens) were genotyped at the same microsatellite loci as were used for the 2014/2015 worker samples described above. However, because in males one locus (BTMS0132) proved monomorphic, data analysis in males was based on a median (range) of 13 (3–13) polymorphic loci.

### (ii) Estimation of levels of diploid male production

Diploid males were assigned as those phenotypic males that were heterozygous at two or more microsatellite loci across two independent rounds of genotyping (as detailed in the electronic supplementary material). All males accepted as diploid (50 of 612 males genotyped) were heterozygous at a mean (range) of 5.5 (2–9) loci.

The diploid male data were then used to produce three estimates of the frequency of colonies exhibiting DMP: (i) from the field-collected colonies; (ii) from the colonies reared from field-collected queens; and (iii) from the latter colonies after correcting for sampling error arising when males were sampled for genotyping (see the electronic supplementary material).

### (iii) Estimation of allelic diversity at the sex-determining locus

By applying the formula of Adams *et al.* [44] and an approach accounting for facultative polyandry, our data on the frequency of colonies exhibiting DMP and previous data on the frequency distribution of queen mating frequencies in the main study *B. hypnorum* population [37] were used to estimate the number of alleles at the sex-determining locus (as detailed in the electronic supplementary material). Boundary values were calculated by assuming either (i) 100% single mating of queens or (ii) 50% double mating and 50% triple mating of queens, as increasing levels of polyandry have the greatest effect on the estimated number of alleles at this locus (see the electronic supplementary material).

All data analyses were carried out using R, unless stated otherwise.

## 3. Results

## (a) Genetic diversity and bottleneck analyses

Across the 645 workers and 14 microsatellite loci, the median number (range) of alleles per locus was 5.0 (3–11), mean allelic richness was 5.9, and mean observed and expected heterozygosity were, respectively, 0.48 and 0.51.

The sign test found no evidence of a recent bottleneck, as the number of loci that had an excess of heterozygosity (7.0 of 14 loci) was not significantly different from that expected (7.5 of 14 loci; $p = 0.49$).

The M-ratio (mean ± standard error) was $0.38 \pm 0.05$. This fell within the range of the mean M-ratios for the five reference *Bombus* species (figure 1), and overall, across these species plus *B. hypnorum*, there was no significant difference in mean M-ratio (ANOVA, $F_{5,31.9} = 1.95$, $p = 0.11$). Unexpectedly, all *Bombus* species exhibited M-ratios under 0.7 (figure 1), indicating some support for historical population reductions in all these populations. However, the overall conclusion from both the sign test and the M-ratio analysis was that there was no strong evidence for a recent severe genetic bottleneck in the UK *B. hypnorum* population relative to UK populations of other *Bombus* species.

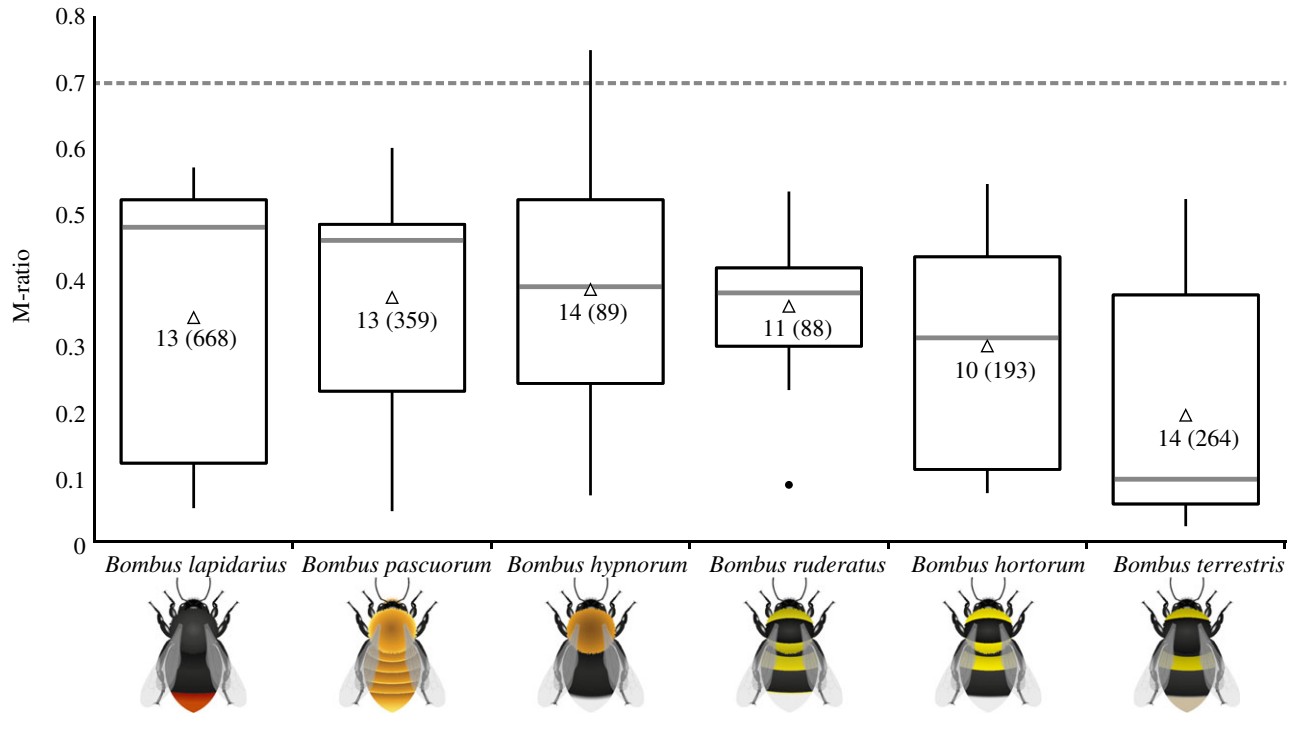

**Figure 1.** M-ratios across microsatellite loci for the main UK study population of *Bombus hypnorum* (current study) and for single UK populations of five other UK *Bombus* species (calculated using data from Dreier *et al.* [54]) in order (left to right) of decreasing medians. Thick horizontal bar, median; box, interquartile range (IQR); whiskers, range (not including outliers); filled circles, outliers, defined as points more than 1.5 IQR below lower quartile; white triangles, means. Numbers within the boxes, number of microsatellite loci and, in parentheses, number of unrelated workers used to calculate the M-ratios for each species. Dashed line at 0.7, M-ratio threshold below which a historical population reduction is hypothesized to have occurred. (Online version in colour.)

## (b) Diploid male production and allelic diversity at the sex-determining locus

### (i) Levels of diploid male production

Across the 612 males and 13 microsatellite loci, the median (range) number of alleles per locus was 5.0 (3–8), and mean allelic richness was 5.3 (electronic supplementary material, table S6), broadly matching the results from the worker genotypes.

All 20 field-collected colonies produced diploid offspring (electronic supplementary material, table S1) and diploid male pupae were found in one of them (5%), accounting for 4.5% of all genotyped male pupae (figure 2a).

Genotyping of adult males from the 37 colonies reared from field-collected queens showed that, of the 32 colonies that produced diploid offspring (including one male-only producing colony that produced a diploid male), five (15.6%) produced diploid males (electronic supplementary material, tables S3–S5). These five DMP colonies comprised four of 12 (33.3%) first-brood male producing colonies and one of 16 (6.3%) late male producing colonies (figure 2b,c). At the level of individual males, four of 25 (16.0%) first-brood males and 29 of 207 (14.0%) late males were found to be diploid (electronic supplementary material, table S3), with diploid males accounting for 14.2% of all genotyped males.

Of the 26 colonies reared from field-collected queens, producing diploids and retained after correcting for sampling error, four (15.4%) produced diploid males (electronic supplementary material, tables S3–S5).

Overall, therefore, 15.4–15.6% of colonies reared from field-collected queens produced diploid males. Moreover, given that

male diploidy was genetically confirmed in only 33.3% of first-brood male producing colonies and 16.0% of first-brood males, first-brood male production was a poor indicator of DMP, as previously it has been assumed that 100% of first-brood male producing colonies would be diploid male producing colonies and 100% of first-brood males would be diploid males.

### (ii) Allelic diversity at the sex-determining locus

Estimates of the proportion of colonies exhibiting DMP ($D$) for field-collected colonies, colonies reared from field-collected queens and colonies reared from field-collected queens after correcting for sampling error were 1/20, 5/32 and 4/26, respectively, or 0.05, 0.156 and 0.154, respectively (as above). Combined with published data on the relative frequencies of singly, doubly and triply mated queens in the main study population [37], these yielded estimates of the frequency of matched mating ($p$) of 0.029, 0.094 and 0.093, respectively, and hence estimates of the number of alleles at the sex-determining locus ($N$) of 69.0, 21.3 and 21.5, respectively (as detailed in the electronic supplementary material). Taking the last estimate to be the most accurate, and adding the calculated boundary values, led to an estimated number (with boundary values) of alleles at the sex-determining locus in the UK *B. hypnorum* population of 21.5 (13.0–30.8) alleles.

## 4. Discussion

We analysed genotypic data from a UK population of the Tree Bumblebee (*B. hypnorum*) to discriminate between two hypotheses for colonization of the UK by this range-expanding

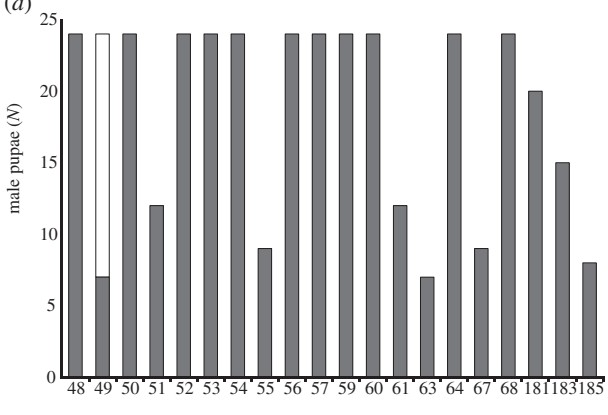

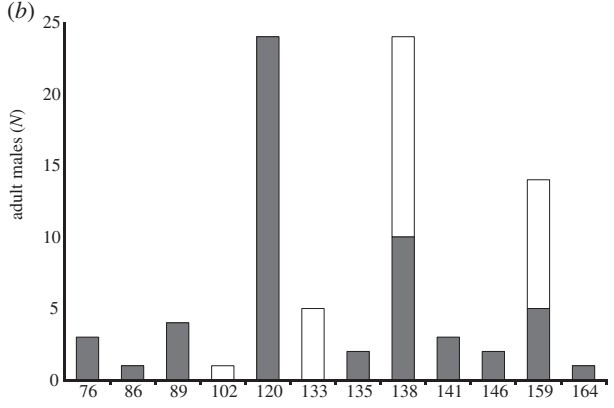

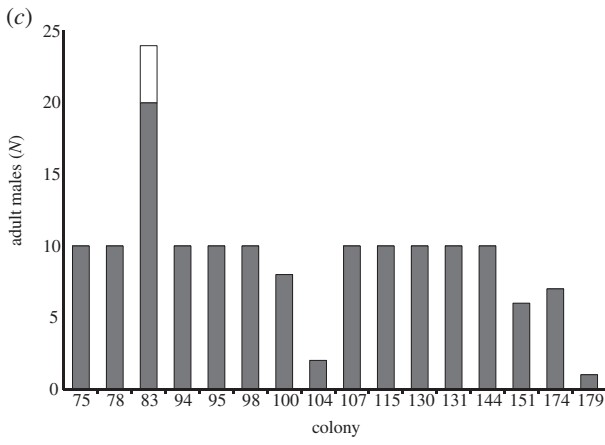

**Figure 2.** Numbers of haploid (dark grey bars) and diploid (white bars) pupal or adult males verified by genotyping at up to 13 microsatellite loci in: (*a*) 20 mature, field-collected *Bombus hypnorum* colonies; (*b*) 12 first-brood male producing *B. hypnorum* colonies reared from field-collected queens; and (*c*) 16 late male producing *B. hypnorum* colonies reared from field-collected queens. Numbers on *x* axes are individual colony identification numbers. (Online version in colour.)

species. The bottleneck hypothesis (small founding population in a single, chance colonization event) predicted low genetic diversity, a recent severe genetic bottleneck and high levels of DMP. The gene flow hypothesis (larger founding population and subsequent continued immigration from continental European populations) predicted high genetic diversity, no genetic bottleneck and low levels of DMP. Our results showed relatively high allele numbers at microsatellite loci and (as estimated from DMP levels) at the sex-determining locus, no evidence of a recent severe genetic bottleneck and relatively low levels of DMP. Therefore, they do not support the bottleneck hypothesis and instead support the gene flow hypothesis. Consequently, although *B. hypnorum* has rapidly and successfully expanded its

range, it does not represent an example of the genetic paradox of invasion.

### (a) Genetic diversity and bottleneck analyses

Previous evidence suggested that the founding UK *B. hypnorum* population numbered as few as one or two multiply-mated queens [38]. However, two queens mated with a mean 1.7 males each [37] would yield a maximum of 7.4 alleles at any locus. This number of alleles is lower than the maximum numbers of alleles found at microsatellite loci in the combined data from workers in Crowther *et al.* [37] and males in the current study (e.g. 11, 9, 9 and 8 alleles at the loci BTMS0125, B10, BL03 and BTERN02, respectively). Therefore, the allele number data do not support the UK *B. hypnorum* population having been founded in a single event by as few as two multiply-mated queens.

Expected heterozygosity and allelic richness at microsatellite loci in the study *B. hypnorum* population ($H_e = 0.51$, $A_R = 5.9$) were both higher than values reported in Belgian and Estonian *B. hypnorum* populations for which comparable data exist (Belgium: $H_e = 0.37$–$0.39$, $A_R = 1.94$–$2.03$; Estonia: $H_e = 0.33$, $A_R = 1.94$) [55], while Huml *et al.* [47] found similar sequence-level diversity between UK *B. hypnorum* populations and a French population. In addition, expected heterozygosity in the study *B. hypnorum* population was, on average, intermediate between values found by previous studies in common, established, and widespread European *Bombus* species and scarce, range-restricted, and/or declining species (electronic supplementary material, table S8). Specifically, it was lower than values for 7/8 populations of common species, and higher than those for 7/8 populations of scarce declining species (electronic supplementary material, table S8). Combined, these genetic diversity comparisons again support the lack of a severe bottleneck in the colonization of the UK by *B. hypnorum*.

Given that *B. hypnorum* is rapidly expanding its range across the UK, and possibly recently did so across north-western Europe [35,45,46], one might expect some loss of genetic diversity to occur, since the leading edge of a dispersal front is subject to a loss of alleles and heterozygosity as it moves further from the source population [56]. This is especially true under leptokurtic dispersal [56], where small numbers of long-distance dispersers found new sub-populations, as appears likely to be the case for *B. hypnorum* queens in the UK [48]. The fact that the UK *B. hypnorum* study population shows a level of expected heterozygosity lower than values found in common European *Bombus* species provisionally supports this idea (electronic supplementary material, table S8). However, the idea requires full testing by sampling an extensive series of *B. hypnorum* populations across the UK and continental Europe.

*Bombus* species are annual insects that typically undergo one generation per year. Some, including *B. hypnorum*, have been suggested to exhibit facultative bivoltinism, i.e. two colony cycles (and hence two generations) per year [57], but there is little evidence that the second generation produces many new queens. Therefore, an estimated minimum of 14–15 generations passed between *B. hypnorum*'s colonization of the UK in or shortly before 2001 and the worker sampling in the current study. Based on previous power analyses [52], our sign test was sufficiently powerful to have detected a bottleneck 0.25 *N* to 2.5 *N* generations after the initial bottleneck, where *N* equals the founding population size (number of diploid individuals) immediately after a

putative bottleneck [52]. Therefore, the sign test should have detected a bottleneck of 6–60 diploid individuals (15 generations/2.5 = 6, and 15 generations/0.25 = 60). As the sign test found no evidence of a bottleneck, it is therefore unlikely that the founding population size was smaller than 60 diploid individuals (120 haploid genomes). In haplodiploid eusocial Hymenoptera, this is equivalent to either 40 singly mated or 30 doubly mated queens. These calculations again reject the bottleneck hypothesis and instead support the gene flow hypothesis, since, if colonization of the UK was part of a large-scale westward range expansion in *B. hypnorum*, one would expect the number of immigrating queens in the founding year and in each subsequent year to have been relatively high.

The M-ratio analysis found no evidence for a level of historical population reduction in *B. hypnorum* greater than the levels found in five other UK *Bombus* species (figure 1). Unexpectedly, M-ratios for all tested *Bombus* species, including *B. hypnorum*, fell below the mutation–drift equilibrium threshold of 0.7 (figure 1), suggesting that the sampled populations of all species have undergone some degree of historical population reduction, potentially up to approximately 100 generations ago [53]. Reasons for this are unknown, as data on *Bombus* distribution and abundances within the UK are mostly limited to recent decades [58]. Conceivably, the finding points to former population reductions in all *Bombus* species at lowland agricultural sites [54], perhaps associated with historical changes in agricultural practices. Regardless, the conclusion remains that, in the recent UK *B. hypnorum* population, there was no evidence of a bottleneck more intense than in populations of long-established UK *Bombus* species.

## (b) Diploid male production and allelic diversity at the sex-determining locus

In the study population of *B. hypnorum*, frequencies of DMP colonies were 5% for mature, field-collected colonies and 15.4–15.6% for colonies reared from field-collected queens. Given observed levels of facultative polyandry in the main study population [37], these values equated to matched mating frequencies of 2.9% and 9.3–9.4%, respectively. Genetic studies of *B. hypnorum* from continental Europe found no diploid males. However, in these studies, colony sample sizes were lower, i.e. 13 colonies [42] or 10 colonies [43], such that comparisons with the UK data return no statistically significant difference in DMP frequencies between UK and continental Europe populations (totals of 6/52 DMP colonies versus 0/23 DMP colonies, respectively: Fisher's exact $p = 0.169$). Therefore, at most, DMP frequency is only moderately higher in the UK than in continental European *B. hypnorum* populations. This is again consistent with the gene flow hypothesis, qualified by some loss of genetic diversity having occurred at the dispersal front represented by the UK *B. hypnorum* population.

Previously, a frequency of DMP colonies of 23.1% (3 of 13 colonies) was reported in a UK *B. hypnorum* population [38], but this was based on the assumption that all first-brood male producing colonies exhibited DMP. However, our findings demonstrate that only 33.3% of such colonies exhibit DMP (figure 2b). If this was the case in the first-brood male producing colonies in the previous study [38], and 6.3% of the other colonies in that study exhibited DMP (as in the current study), then the frequency of DMP colonies in the previous study [38] can be estimated as 12.5% (from [(0.333 × 3) + (0.063 × 10)]/13 = 0.125), consistent with the values estimated in the current study.

High DMP frequencies are characteristic of populations of eusocial Hymenoptera known from other evidence to have suffered severe bottlenecks during the colonization of new ranges. For example, the French *V. velutina* population was founded by a single polyandrous queen [23], and DMP is observed in 48.3% of field-collected nests [20]. Similarly, the Tasmanian *B. terrestris* population was founded by two monandrous queens [24], and DMP was inferred in 50% of colonies reared from field-collected queens [19]. Correspondingly, the relatively lower levels of DMP observed in the UK *B. hypnorum* population do not support the occurrence of a severe genetic bottleneck.

The estimated number of alleles at the sex-determining locus in the UK *B. hypnorum* population (with boundary values calculated by assuming 100% single mating or 50% double mating and 50% triple mating of queens) was 21.5 (13.0–30.8) alleles. This estimate is consistent with the conclusion from the microsatellite data that the UK *B. hypnorum* population has not undergone a severe bottleneck, with even the lower bound exceeding the previous estimate of this number [38] by over threefold. Based on genetic assays of DMP, a total of eight alleles were estimated at the sex-determining locus in *B. florilegus*, a species that has undergone a severe range contraction across its Japanese range [59]. The contrast between this value and the value estimated from the UK *B. hypnorum* population is again consistent with the lack of an extreme reduction in genetic diversity in the latter population.

Combining the genetic and productivity data in the *B. hypnorum* colonies reared from field-collected queens suggests that DMP decreased colony productivity (DMP colonies ($n = 5$): mean $n_{workers} = 18$, mean $n_{gynes} = 0$, mean $n_{males} = 15$; non-DMP colonies ($n = 32$): mean $n_{workers} = 34$, mean $n_{gynes} = 6$, mean $n_{males} = 43$). This suggestion is in line with previous findings in ants and bees [15–18], and exemplifies the fitness costs of matched matings in bumblebees. Such reductions in colony productivity may account for the lower DMP frequencies observed in mature, field-collected colonies than in queen-reared colonies (figure 2), with smaller colonies being less likely to survive and be available for sampling, and support the assumption of sampling and/or survival bias in estimating DMP frequencies from field-collected colonies.

In conclusion, genetic data from the UK *B. hypnorum* population showed relatively high genetic diversity, no evidence of a recent severe genetic bottleneck and low levels of DMP, matching predictions from the gene flow hypothesis. Hence, colonization of the UK by *B. hypnorum* does not represent an example of the genetic paradox of invasion [5,38] or an example of a eusocial Hymenopteran achieving rapid range expansion despite high levels of DMP [19,20]. Alongside evidence of *B. hypnorum* undergoing a recent westward range expansion within Europe [35,45,46], our findings suggest that this species may resemble other invertebrate taxa that have recently expanded their ranges at a continental scale. Examples in Europe include the wasp spider (*Argiope bruennichi*) [60] and the dainty damselfly (*Coenagrion scitulum*) [61]. In such cases, the central genetic phenomena are progressive loss of genetic diversity across an invasion front and its consequent impact on adaptability [56,62], along with the

genetic processes, if any, that trigger range expansions away from the source population. Whether the above-mentioned genetic phenomena impact *B. hypnorum*'s distinctive pollinating role in the UK [36] remains to be discovered.

Data accessibility. Colony/queen collection, queen rearing and colony productivity data are available in the electronic supplementary material. Genotyping data are available from the Dryad Digital Repository: https://doi.org/10.5061/dryad.t76hdr7zv [63]. Code for the bottleneck and diploid male analyses is available from GitHub at: https://github.com/RBrock94/TreeBumblebee_Bottleneck.

Authors' contributions. D.S.R., C.C., M.I.T. and A.F.G.B. conceived and supervised the study; R.E.B., L.P.C., D.S.R., C.C., M.I.T. and A.F.G.B. designed the study; R.E.B. and L.P.C. conducted the field and laboratory work; D.S.R., D.J.W. and M.I.T. provided laboratory and genetic advice; R.E.B. and L.P.C. performed data analyses; A.F.G.B. produced the model to estimate allelic diversity at the sex-determining locus; R.E.B. and A.F.G.B. wrote the original draft of the manuscript, with input from L.P.C.; and all authors reviewed the final draft and gave approval for publication.

Competing interests. The authors declare no competing interests.

Funding. This work was supported by a UK Natural Environment Research Council PhD studentship held by L.P.C., in a CASE partnership with the UK Centre for Ecology and Hydrology, Wallingford (NE/L50158X/1), and a UK Natural Environment Research Council PhD studentship held by R.E.B. within the EnvEast Doctoral Training Partnership (NE/L002582/1).

Acknowledgements. We thank Kerry Blair for assistance with pupal genotyping; Mark Brown and Arran Folly for kindly hosting R.E.B.'s visit to the Brown laboratory at Royal Holloway, University of London and providing generous advice on rearing *B. hypnorum*; Tracey Chapman for comments on earlier drafts of the manuscript; Richard Goram (John Innes Centre, Norwich) for genotyping services; Vanessa Huml and Mairi Knight for sharing unpublished work; Zoe Keeble for assistance with nest collections; and the Norfolk Beekeepers' Association and members of the public for notifying us of *B. hypnorum* nests and allowing their collection.

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
