## [Reviewer comments · Proceedings of the Royal Society B: Biological Sciences]

Review History

RSPB-2020-2639.R0 (Original submission)

Review form: Reviewer 1

Recommendation

Accept with minor revision (please list in comments)

Scientific importance: Is the manuscript an original and important contribution to its field?

Good

General interest: Is the paper of sufficient general interest?

Acceptable

Quality of the paper: Is the overall quality of the paper suitable?

Good

Is the length of the paper justified?

Yes

Should the paper be seen by a specialist statistical reviewer?

No

Do you have any concerns about statistical analyses in this paper? If so, please specify them explicitly in your report.

No

It is a condition of publication that authors make their supporting data, code and materials available - either as supplementary material or hosted in an external repository. Please rate, if applicable, the supporting data on the following criteria.

Is it accessible?

Yes

Is it clear?

Yes

Is it adequate?

Yes

Do you have any ethical concerns with this paper?

No

Comments to the Author

This is an interesting, well-written manuscript where the authors set to distinguish between two hypotheses for the mode of UK colonisation by the tree bumblebee. It was previously published that the species had suffered a severe bottleneck when colonising the UK, but this conclusion was reached without genetic data for diploid male production (DMP) or accounting for facultative polyandry, two phenomena of importance in haplodiploid species. The authors expand a previous set of microsatellite loci that were screened for workers by adding males and they estimate the frequency of diploid males and allelic diversity at the sex-determining locus for this species. The paper nicely shows that the colonisation of the UK by tree bumblebee does not represent an example of the genetic paradox of invasion.

I only have very minor comments.

The M-ratio analyses and figure are a nice addition to the paper. However, even the authors are not certain about how to interpret the M-ratios values in relation to some historical population reduction of the bumblebee species compared. I wonder if comparing M-ratios between different populations of the tree bumblebee in the UK (recent expanded range) vs France or Belgium (native range) should shed better light to the interpretations of the M-ratio values. Figure 1. Instead of ordering the species alphabetically why not show them in increasing order of M-ratio?

Please replace 'bottlenecking' for 'bottleneck' throughout the manuscript.

The Huml et al. paper cited here as submitted 'somewhere' seems to answer part of this study aims; colonisation of the UK by *B. hypnorum* represented part of an ongoing, large-scale range expansion, with a larger founding population and subsequent continued immigration from mainland Europe, using RAD-Seq. Maybe focusing your manuscript on the estimations of the frequency of diploid males and allelic diversity at the sex-determining locus might make the two papers stand alone and reduce the overlap? Just a thought.

Review form: Reviewer 2

Recommendation

Accept with minor revision (please list in comments)

Scientific importance: Is the manuscript an original and important contribution to its field?

Excellent

General interest: Is the paper of sufficient general interest?

Excellent

Quality of the paper: Is the overall quality of the paper suitable?

Excellent

Is the length of the paper justified?

Yes

Should the paper be seen by a specialist statistical reviewer?

No

Do you have any concerns about statistical analyses in this paper? If so, please specify them explicitly in your report.

No

It is a condition of publication that authors make their supporting data, code and materials available - either as supplementary material or hosted in an external repository. Please rate, if applicable, the supporting data on the following criteria.

Is it accessible?

Yes

Is it clear?

Yes

Is it adequate?

Yes

Do you have any ethical concerns with this paper?

No

Comments to the Author

The manuscript investigates whether the invasive UK population of the tree bumblebee *Bombus hypnorum* has undergone a genetic bottleneck, as is commonly believed, or exhibits continued gene flow from continental Europe. It has generally been considered that the invasion originated with just a couple of queens and that its success in spite of this genetic bottleneck thus represents a classic example of the genetic paradox of invasion. In this manuscript, the authors combine genetic diversity data with data on the production of diploid males (an indicator of a genetic bottleneck) to provide strong evidence that the UK population exhibits continued gene flow. This is important for our understanding both of the successful invasion history of the tree bumblebee in the context of declines in many other pollinator species, and of invasion biology in general.

The study is well conducted, bringing together multiple lines of evidence to support the important conclusion, and the manuscript is well written. I have only a few minor comments:

1. The Discussion is perhaps rather longer than is strictly necessary and could be shortened.
2. Lines 288-303. Many heterozygosity values are listed here for other bumblebees. These are very useful for comparison, but it might be better to put these in a figure, together with the value you found for *B. hypnorum*. Putting them in a figure would both make them easier to digest and reduce the length of the Discussion.

3. Line 189. It's stated here that the analysis is based on a median of 13 microsat loci, with a range of 3-13. Is the median value correct?
4. Line 221. Here or earlier it might be helpful for the reader's understanding of the M-ratio results to say what ratio would be indicative of a bottleneck.
5. Line 287. It would aid interpretation of this conclusion if an estimate was added of how many queens would be the minimum necessary to explain the number of alleles found. I'd guess that it is still only a small number and that this may therefore not be a particularly important finding.
6. Line 381. This paragraph could be reduced. The point about high frequency of DMP being an indicator of genetic bottleneck is probably best made (succinctly) in the Introduction.

Decision letter (RSPB-2020-2639.R0)

29-Dec-2020

Dear Mr Brock

I am pleased to inform you that your manuscript RSPB-2020-2639 entitled "No severe genetic bottleneck in a rapidly range-expanding bumblebee pollinator" has been accepted for publication in Proceedings B.

The referee(s) have recommended publication, but also suggest some minor revisions to your manuscript. Therefore, I invite you to respond to the referee(s)' comments and revise your manuscript. Because the schedule for publication is very tight, it is a condition of publication that you submit the revised version of your manuscript within 7 days. If you do not think you will be able to meet this date please let us know.

- 1) A text file of the manuscript (doc, txt, rtf or tex), including the references, tables (including captions) and figure captions. Please remove any tracked changes from the text before submission. PDF files are not an accepted format for the "Main Document".
- 2) A separate electronic file of each figure (tiff, EPS or print-quality PDF preferred). The format should be produced directly from original creation package, or original software format. PowerPoint files are not accepted.
- 3) Electronic supplementary material: this should be contained in a separate file and where possible, all ESM should be combined into a single file. All supplementary materials

accompanying an accepted article will be treated as in their final form. They will be published alongside the paper on the journal website and posted on the online figshare repository. Files on figshare will be made available approximately one week before the accompanying article so that the supplementary material can be attributed a unique DOI.

It is a condition of publication that data supporting your paper are made available either in the electronic supplementary material or through an appropriate repository. Please see our Data Sharing Policies <https://royalsociety.org/journals/authors/author-guidelines/#data>.

Sincerely,

Dr Sasha Dall

Associate Editor

Board Member: 1

Comments to Author:

Thank you for your patience during what has been an unfortunately extended review process. Two expert reviewers have now submitted their evaluations of your manuscript, and they largely concur in their suggestions for how to improve it:

1. Revise your text to better explain or "teach" to the reader how to interpret M-ratio results, including expected values associated with null and preferred hypotheses.
2. Clarify the relationship between this manuscript and the Huml et al manuscript in submission. Reviewers are concerned with potential overlap between these two manuscripts and, hence, the novelty of the present paper. Manuscripts have been rejected for similar reasons.

I would like to see a revision that addresses each point raised by the reviewers while working to streamline the discussion.

Reviewer(s)' Comments to Author:

Referee: 1

Comments to the Author(s)

This is an interesting, well-written manuscript where the authors set to distinguish between two hypotheses for the mode of UK colonisation by the tree bumblebee. It was previously published that the species had suffered a severe bottleneck when colonising the UK, but this conclusion was reached without genetic data for diploid male production (DMP) or accounting for facultative polyandry, two phenomena of importance in haplodiploid species. The authors expand a previous set of microsatellite loci that were screened for workers by adding males and they estimate the frequency of diploid males and allelic diversity at the sex-determining locus for this species. The paper nicely shows that the colonisation of the UK by tree bumblebee does not represent an example of the genetic paradox of invasion.

I only have very minor comments.

The M-ratio analyses and figure are a nice addition to the paper. However, even the authors are not certain about how to interpret the M-ratios values in relation to some some historical population reduction of the bumblebee species compared. I wonder if comparing M-ratios between different populations of the tree bumblebee in the UK (recent expanded range) vs France or Belgium (native range) should shed better light to the interpretations of the M-ratio values. Figure 1. Instead of ordering the species alphabetically why not show them in increasing order of M-ratio?

Please replace 'bottlenecking' for 'bottleneck' throughout the manuscript.

The Huml et al. paper cited here as submitted 'somewhere' seems to answer part of this study aims; colonisation of the UK by *B. hypnorum* represented part of an ongoing, large-scale range expansion, with a larger founding population and subsequent continued immigration from mainland Europe, using RAD-Seq. Maybe focusing your manuscript on the estimations of the frequency of diploid males and allelic diversity at the sex-determining locus might make the two papers stand alone and reduce the overlap? Just a thought.

Referee: 2

Comments to the Author(s)

The manuscript investigates whether the invasive UK population of the tree bumblebee *Bombus hypnorum* has undergone a genetic bottleneck, as is commonly believed, or exhibits continued gene flow from continental Europe. It has generally been considered that the invasion originated with just a couple of queens and that its success in spite of this genetic bottleneck thus represents a classic example of the genetic paradox of invasion. In this manuscript, the authors combine genetic diversity data with data on the production of diploid males (an indicator of a genetic bottleneck) to provide strong evidence that the UK population exhibits continued gene flow. This is important for our understanding both of the successful invasion history of the tree bumblebee in the context of declines in many other pollinator species, and of invasion biology in general.

The study is well conducted, bringing together multiple lines of evidence to support the important conclusion, and the manuscript is well written. I have only a few minor comments:

1. The Discussion is perhaps rather longer than is strictly necessary and could be shortened.

2. Lines 288-303. Many heterozygosity values are listed here for other bumblebees. These are very useful for comparison, but it might be better to put these in a figure, together with the value you found for *B. hypnorum*. Putting them in a figure would both make them easier to digest and reduce the length of the Discussion.
3. Line 189. It's stated here that the analysis is based on a median of 13 microsat loci, with a range of 3-13. Is the median value correct?
4. Line 221. Here or earlier it might be helpful for the reader's understanding of the M-ratio results to say what ratio would be indicative of a bottleneck.
5. Line 287. It would aid interpretation of this conclusion if an estimate was added of how many queens would be the minimum necessary to explain the number of alleles found. I'd guess that it is still only a small number and that this may therefore not be a particularly important finding.
6. Line 381. This paragraph could be reduced. The point about high frequency of DMP being an indicator of genetic bottleneck is probably best made (succinctly) in the Introduction.

Author's Response to Decision Letter for (RSPB-2020-2639.R0)

See Appendix A.

Decision letter (RSPB-2020-2639.R1)

18-Jan-2021

Dear Mr Brock

I am pleased to inform you that your manuscript entitled "No severe genetic bottleneck in a rapidly range-expanding bumblebee pollinator" has been accepted for publication in Proceedings B.

Your article has been estimated as being 9 pages long. Our Production Office will be able to confirm the exact length at proof stage.

Open Access

You are invited to opt for Open Access, making your freely available to all as soon as it is ready for publication under a CCBY licence. Our article processing charge for Open Access is £1700. Corresponding authors from member institutions

Paper charges

Sincerely,
Proceedings B
<mailto:proceedingsb@royalsociety.org>

Appendix A

Proceedings B Manuscript ID RSPB-2020-2639: Response to referees

Ryan E. Brock, Liam P. Crowther, David J. Wright, David S. Richardson, Claire Carvell, Martin I. Taylor & Andrew F. G. Bourke. *No severe genetic bottleneck in a rapidly range-expanding bumblebee pollinator*

We are grateful for the positive comments received from the associate editor and reviewers and would like to thank them for their time spent evaluating our submission. We have revised the manuscript according to all the comments received and have streamlined the Discussion to be more concise.

In our point-by-point response (below), editor/reviewer comments are in *blue italics* and our author responses are in black Roman type. We have numbered all referees' points for ease of reference. Line numbers in our responses refer to the revised manuscript unless otherwise stated. As requested, a copy of the revised manuscript with tracked changes can be found at the end of the responses.

Associate Editor, Comments to Author:

Thank you for your patience during what has been an unfortunately extended review process. Two expert reviewers have now submitted their evaluations of your manuscript, and they largely concur in their suggestions for how to improve it:

1. Revise your text to better explain or "teach" to the reader how to interpret M-ratio results, including expected values associated with null and preferred hypotheses.

Author response: We have now revised the text to explain more clearly to the reader how M-ratio results should be interpreted, in terms of both the M-ratio value we would expect a bottlenecked population to exhibit (<0.7) and what we can infer from the comparisons between the *B. hypnorum* M-ratio and those generated for other established UK bumblebee species. Specifically, the following additions have been made to the manuscript:

- Lines 153-155 – Sentence added: *“Under mutation-drift equilibrium, an M-ratio of less than 0.7 can be interpreted as evidence of a historical population reduction, a signal that may persist for more than 100 generations [1].”*
- Lines 161-163 – Sentence added: *“Hence, if *B. hypnorum* experienced a severe genetic bottleneck upon its colonisation of the UK, it would be expected to exhibit a lower M-ratio than the reference *Bombus* species.”*
- Lines 611-619 & Figure 1 – Dashed line added to Figure 1 to show the M-ratio value of 0.7, and figure legend updated as follows (at lines 615-616) – *“Orange dashed line at 0.7, M-ratio threshold below which a historical population reduction is hypothesised to have occurred.”*

2. Clarify the relationship between this manuscript and the Huml et al manuscript in submission. Reviewers are concerned with potential overlap between these two manuscripts

and, hence, the novelty of the present paper. Manuscripts have been rejected for similar reasons.

Author response: This was a point raised by Referee 1. In response, please note that the Huml *et al.* study is an independent study conducted by another group that we have seen in ms through that group kindly sharing it with us (we also shared with them the original submission of the current ms). As the Huml *et al.* study is not yet published, we have cited it as a courtesy and because we think that, in general, it is good for science as a whole if groups working on related issues share information. The Huml *et al.* study uses RAD-seq techniques and, because it included only one non-UK population (a single French population), supports (without conclusively proving) genetic continuity between UK and continental European populations of *B. hypnorum*.

Our study differs, and is novel, through taking the completely unrelated approach of (1) conducting a comparative bottleneck analysis of a UK *B. hypnorum* population and populations of other UK *Bombus* species, and (2) presenting the first genetic measure of diploid male production levels in UK *B. hypnorum*, so permitting a robust calculation of the number of alleles at the sex-determining locus. As argued in our ms, the results from each of these elements reach the same conclusion, i.e. that the UK *B. hypnorum* population has not undergone a severe genetic bottleneck and instead shows features consistent with the gene flow hypothesis. In addition, it is worth noting that the present group was the first to propose the gene flow hypothesis for the UK *B. hypnorum* population (in 2018 in Ref. 48), so the current ms represents a natural development of our group's research in this area.

Overall, therefore, we submit that the current ms and the Huml *et al.* ms are indeed distinct. In addition, we submit that it strengthens a given scientific case, including the current one, when two independent studies with overlapping but not identical aims and different methods reach complementary, mutually supportive conclusions. To reflect these points, we have now added the citation of Ref. 48 to the Introduction (line 117), and added text to the final passage of the Introduction to spell out the novelty of the current study more fully (line 123).

3. I would like to see a revision that addresses each point raised by the reviewers while working to streamline the discussion.

Author response: In the revised ms, we have now addressed each referee's point and streamlined the Discussion. For details, please see our specific responses below.

Referee 1, Comments to the Author(s):

This is an interesting, well-written manuscript where the authors set to distinguish between two hypotheses for the mode of UK colonisation by the tree bumblebee. It was previously published that the species had suffered a severe bottleneck when colonising the UK, but this conclusion was reached without genetic data for diploid male production (DMP) or accounting for facultative polyandry, two phenomena of importance in haplodiploid species. The authors expand a previous set of microsatellite loci that were screened for workers by adding males and they estimate the frequency of diploid males and allelic diversity at the sex-

determining locus for this species. The paper nicely shows that the colonisation of the UK by tree bumblebee does not represent an example of the genetic paradox of invasion.

I only have very minor comments.

1. The M-ratio analyses and figure are a nice addition to the paper. However, even the authors are not certain about how to interpret the M-ratios values in relation to some historical population reduction of the bumblebee species compared. I wonder if comparing M-ratios between different populations of the tree bumblebee in the UK (recent expanded range) vs France or Belgium (native range) should shed better light to the interpretations of the M-ratio values.

Author response: The referee is correct that it is puzzling as to why populations of *Bombus* species long established in the UK should show relatively low M-ratios, and a possible reason why was discussed in the original ms (original ms: lines 349-355; revised ms: lines 337-344). Unfortunately, however, we cannot implement the referee's suggestion because there are no published population-genetic data from continental European populations of *B. hypnorum* that would allow us to carry out M-ratio tests comparable with the one presented in this current study (or indeed to test for gene flow directly). Population-genetic data for continental European *B. hypnorum* populations can be found in Paxton *et al.*, 2001 (*Molecular Ecology*, **10**, 2489-2498), Brown *et al.*, 2003 (*Molecular Ecology*, **12**, 1599-1605), and Maebe *et al.*, 2019 (*Scientific Reports*, **9**, 1-8) for Finnish, Swedish, and Estonian and Belgian populations, respectively. In the case of Paxton *et al.* (2001) and Brown *et al.* (2003), colony number sample sizes (10 and 14 colonies, respectively) are not large enough to produce the numbers of unrelated workers required for reliable M-ratio calculations (sampling one worker per colony), and the workers were not typed at a comparable number of microsatellite loci (4 and 6 loci typed, respectively) relative to the current study. In the case of Maebe *et al.* (2019), worker number sample sizes (3 and 18 for Estonian and Belgian populations, respectively) are very low and once again not large enough to produce the number of unrelated workers required for reliable M-ratio calculation.

We have now also looked at expanding the sample size of populations of long-established UK *Bombus* species with which to compare *B. hypnorum* in the M-ratio analysis in the current study. The existing populations included in the study, which were taken from Dreier *et al.*, 2014 (*Molecular Ecology*, **23**, 3384-3395), were selected for their taxonomic breadth, relative geographic proximity to the *B. hypnorum* sampling site, similar number of loci used for genotyping (range = 10–14), overlap of molecular markers used between the studies, and a worker sampling protocol similar to the one used in the current study (original ESM, lines 89-93). We find that, although there are other published studies presenting population-genetic data for UK *Bombus* species, no others fit all or most of this set of requirements, meaning this comparison is also not feasible. However, the comparison that we already present, with five other *Bombus* populations, already demonstrates that *B. hypnorum* is not an outlier among UK *Bombus* populations in terms of its M-ratio, and hence that its recent colonisation of the UK is not associated with exceptional loss of genetic diversity.

2. Figure 1. Instead of ordering the species alphabetically why not show them in increasing order of M-ratio?

Author response: We have now implemented this by ordering the six species in Figure 1 in the order of high to low median M-ratio. To further aid interpretation of the figure, we have added a line at the M-ratio value of 0.7, to show the threshold below which M-ratio values are indicative of a population bottleneck. We have also updated the legend to reflect these changes (lines 611-619).

3. Please replace 'bottlenecking' for 'bottleneck' throughout the manuscript.

Author response: We have now replaced all uses of ‘bottlenecking’ with ‘bottleneck’ or ‘bottlenecks’ in both the main text and the electronic supplementary material.

*4. The Huml et al. paper cited here as submitted ‘somewhere’ seems to answer part of this study aims; colonisation of the UK by *B. hypnorum* represented part of an ongoing, large-scale range expansion, with a larger founding population and subsequent continued immigration from mainland Europe, using RAD-Seq. Maybe focusing your manuscript on the estimations of the frequency of diploid males and allelic diversity at the sex-determining locus might make the two papers stand alone and reduce the overlap? Just a thought.*

Author response: Please see our response above to the associate editor's second comment.

Referee 2, Comments to the Author(s):

*The manuscript investigates whether the invasive UK population of the tree bumblebee *Bombus hypnorum* has undergone a genetic bottleneck, as is commonly believed, or exhibits continued gene flow from continental Europe. It has generally been considered that the invasion originated with just a couple of queens and that its success in spite of this genetic bottleneck thus represents a classic example of the genetic paradox of invasion. In this manuscript, the authors combine genetic diversity data with data on the production of diploid males (an indicator of a genetic bottleneck) to provide strong evidence that the UK population exhibits continued gene flow. This is important for our understanding both of the successful invasion history of the tree bumblebee in the context of declines in many other pollinator species, and of invasion biology in general.*

The study is well conducted, bringing together multiple lines of evidence to support the important conclusion, and the manuscript is well written. I have only a few minor comments:

1. The Discussion is perhaps rather longer than is strictly necessary and could be shortened.

Author response: We have now shortened the Discussion in accordance with the referee's request, decreasing its total length from 2,244 words to 1,822 words (i.e. by 19%).

*2. Lines 288-303. Many heterozygosity values are listed here for other bumblebees. These are very useful for comparison, but it might be better to put these in a figure, together with the value you found for *B. hypnorum*. Putting them in a figure would both make them easier to digest and reduce the length of the Discussion.*

Author response: As suggested, we have now added a new table (in the electronic supplementary material) to present these values (Table S8; ESM lines 604-612) and this has also helped reduce the length of the corresponding passage in the Discussion (lines 298-303).

3. Line 189. It's stated here that the analysis is based on a median of 13 microsat loci, with a range of 3-13. Is the median value correct?

Author response: Most males (N = 413, representing 67.5% of all genotyped males) were successfully genotyped at all 13 loci and so the median of 13 genotyped loci is correct. To clarify this to the reader, we have now added to the text the point that 67.5% of males were successfully genotyped at all 13 loci (ESM lines 198-199).

4. Line 221. Here or earlier it might be helpful for the reader's understanding of the M-ratio results to say what ratio would be indicative of a bottleneck.

Author response: As indicated in our response to the associate editor, we have now made the following revisions to help the reader's understanding of what M-ratio values can be taken to indicate a bottleneck:

- Lines 153-155 – Sentence added: “*Under mutation-drift equilibrium, an M-ratio of less than 0.7 can be interpreted as evidence of a historical population reduction, a signal that may persist for more than 100 generations [1].*”
- Lines 611-619 & Figure 1 – Dashed line added to Figure 1 to show the M-ratio value of 0.7, and figure legend updated as follows (at lines 615-616) – “*Orange dashed line at 0.7, M-ratio threshold below which a historical population reduction is hypothesised to have occurred.*”

5. Line 287. It would aid interpretation of this conclusion if an estimate was added of how many queens would be the minimum necessary to explain the number of alleles found. I'd guess that it is still only a small number and that this may therefore not be a particularly important finding.

Author response: The minimum number of founding queens necessary to explain the total of 11 alleles at the BTMS0125 locus would be 3, based on $11/3.7 = 3$ (where the 3.7 represents the number of alleles carried by a diploid queen and her 1.7 mates, as in Crowther *et al.*, 2019 (*Ecology and Evolution*, **9**, 986-997)). However, this number in isolation is not very informative, as microsatellites represent just one kind of marker and, for example, we estimated a higher number of alleles (21.5) at the sex-determining locus. The more reliable estimate of founding population size comes from the sign test results, which predict a founding population of either 40 singly-mated or 30 doubly-mated queens (original ms: line 332-335; revised ms: lines 325-330). Hence, and because the aim of the passage under consideration was to show that the previous estimate of two queens founding the UK *B. hypnorum* population was incorrect (lines 285-292), we have not added the $11/3.7$ calculation to the Discussion.

6. Line 381. This paragraph could be reduced. The point about high frequency of DMP being an indicator of genetic bottleneck is probably best made (succinctly) in the Introduction.

Author response: We have now shortened this paragraph (lines 368-375) and added the point about high DMP frequency being an indicator of genetic bottleneck to the Introduction (lines 65-66).